# VPTD: Human Face Video Dataset for Personality Traits Detection

Kenan Kassab [1], Alexey Kashevnik [1,2,*], Alexander Mayatin [3] and Dmitry Zubok [3]

1   St. Petersburg Federal Research Center of the Russian Academy of Sciences (SPC RAS), 199178 St. Petersburg, Russia; kkassab@itmo.ru
2   Institute of Mathematics and Information Technologies, Perozavodsk State University (PetrSU), 185035 Petrozavodsk, Russia
3   Information Technology and Programming Faculty, ITMO University, 197101 St. Petersburg, Russia; mayatin@mail.ifmo.ru (A.M.); zubok@itmo.ru (D.Z.)
*   Correspondence: alexey.kashevnik@gmail.com

**Abstract:** In this paper, we propose a dataset for personality traits detection based on human face videos. Ground truth data have been annotated using the IPIP-50 personality test that every participant is implementing. To collect the dataset, we developed a web-based platform that allows us to acquire spontaneous answers for predefined questions from the respondents. The website allows the participants to record an interactive interview in order to imitate the real-life interview. The dataset includes 38 videos (2 min on average) for people of different races, genders, and ages. In the paper, we propose the top five personality traits calculated based on the test, as well as the top five personality traits calculated by our own developed model that determines this information based on video analysis. We introduced a statistical analysis for the collected dataset, and we also applied a K-means clustering algorithm to cluster the data and present the clustering results.

**Keywords:** human face dataset; OCEAN model; personality traits; Principal Component Analysis





## 1. Introduction

Personality traits play a crucial role in shaping the way individuals interact with the world around them. The OCEAN (Openness, Conscientiousness, Extraversion, Agreeableness, and Neuroticism) model, also known as the big five personality traits, is a widely used model that helps identify and understand these traits. Openness refers to an individual's readiness to try out novel experiences and concepts. Conscientiousness is related to being organized, responsible, and dependable. Extraversion relates to an outgoing and social personality, whereas agreeableness is associated with being cooperative and empathetic. Finally, neuroticism pertains to an individual's tendency to experience negative emotions such as anxiety and stress [1,2]. A previous meta-analytic review examines the relationship between the Big Five personality traits (OCEAN model) and entrepreneurial success. It explores how each trait influences entrepreneurial behaviors, such as opportunity recognition, risk-taking, and innovation. The study provides insights into the personality characteristics that contribute to entrepreneurial success and can inform entrepreneurial selection and development processes [3]. Personality traits strongly affect job performance, and many organizations recognize this impact. Conscientiousness has the greatest positive influence on employee performance, followed by extraversion, openness to experience, and agreeableness. Neuroticism, however, has a negative impact on job performance. Considering these traits can guide organizations in making effective decisions regarding employee selection and management strategies [4]. Studies show that individuals who score high in extraversion tend to be better suited for sales positions, as they enjoy social interaction and are comfortable engaging with others. Those with high levels of conscientiousness are typically reliable and detail-oriented, which can be beneficial in a sales role that involves

careful planning and attention to detail. Individuals with low levels of neuroticism may handle the high-pressure nature of sales more effectively [5].

Recently, with the advent of machine learning and computer vision techniques, researchers have explored the possibility of estimating personality traits using video-based datasets. The authors of [6] presented a technique for identifying personality traits by extracting crucial facial landmarks. They employed dimensionality reduction to reduce the feature space and utilized Support Vector Regression (SVR) to build a five-dimensional prediction model using these landmarks. In another study [7], they introduced a multi-modal approach to estimate personality cues. They proposed a method that combines Convolutional Neural Network (CNN) for feature extraction and Long Short-Term Memory (LSTM) for recognizing observable personality traits. Additionally, they created a fusion model by integrating four sub-networks focused on feature-based recognition, namely ambient, facial, audio, and transcription.

In this paper, we provide a video-based dataset for personality trait estimation. The dataset was collected from different participants with different nationalities, genders, and races. The participants used our website to record video interviews answering general questions and custom questions related to the sales field. They also took a personality test to assess their personality traits. The collected dataset could be utilized for many studies related to personality analysis, understanding the relationship between the apparent facile traits and personality, etc.

The collected dataset consists of various samples, primarily composed of students who possess a wide range of personality traits and sales experience. The final dataset comprises 38 self-video interviews, each lasting an average of 120 s. The dataset contains valuable information related to the sales field, where the participants answered interview questions related to sales. It also has the participant's self-estimation of their sales abilities. This makes our dataset unique and new because no such available dataset related to sales estimation has been collected before. We used the collected dataset to study and analyze human ability to work in sales based on personality traits. In this case, such a system can determine non-contactless human ability for sales based on an RGB camera. The scientific novelty of the paper can be summarized as follows:

- Data collection methodology that is suitable for collecting a dataset related to personality traits and soft-skills estimation.
- We presented a pipeline for data analysis that includes our findings related to sales manager clusterization.
- Providing a unique open dataset related to sales estimation that contains personality traits estimation for the participants.

The rest of the paper is organized as follows. Section 2 represents an overview of datasets related to our work. Section 3 talks about the methods and tools we developed to collect the dataset. It also delves into describing and analyzing the dataset. Section 4 talks about experiments we tested to evaluate the dataset. Section 5 discusses the limitations of the study and Section 6 concludes the paper.

## 2. Related Work

In the domains of psychology and social sciences, a great deal of research has been conducted on the Big Five personality traits. In recent years, scientists have used video-based datasets to investigate the connection between personality characteristics and behavior. In this related work, we will explore some of the video-based datasets that have been used to study personality traits according to the OCEAN model.

A large-scale video collection for study on the visual detection of human activities and interactions is the ChaLearn Looking at Humans UDIVA (Understanding Dyadic Interactions from Video and Audio signals) v0.5 dataset [8]. The dataset was unveiled as a

part of the ChaLearn Looking at People (LAP) competition series at the International Conference on Computer Vision (ICCV 2021). The UDIVA dataset consists of 90.5 h of recordings of dyadic interactions involving 147 voluntarily participating individuals from 22 different nations, ranging in age from 4 to 84. The bulk of the participants, who identified as white, were students. One-hundred and eighty-eight dyadic sessions, with an average of 2.5 sessions per participant, were conducted with the participants. Technically, six HD tripod-mounted cameras ($1280 \times 720$ pixels, 25 frames per second), one lapel microphone for each participant, and an omnidirectional microphone on the table were used to collect the data. The personality traits described by the OCEAN model were received from the self-report questionnaire BFI-2 (Big Five Inventory–2 uses 60 items to assess the Big Five personality domains) [9].

The Chalearn First Impression Looking at People (CVPR'17) dataset is a substantial video dataset created for study on the automated evaluation of first impressions made by humans [10]. The dataset was developed as a component of the ChaLearn Looking at People challenge series, and the second version (V2) of the dataset was made available in 2018. Almost 10,000 videos of people introducing themselves and carrying out a quick activity, such as doing a puzzle or reading a text, are included in the First Impressions V2 dataset. The videos were taken from over 3000 separate high-definition (HD) YouTube videos of individuals speaking in front of a camera. The videos include people of all ages, genders, and nationalities. The videos were labeled with personality traits variables. To create the labels, the Amazon Mechanical Turk (AMT) was utilized. The personality traits by the OCEAN model were taken into consideration. As a result, each clip has ground-truth labels for these five qualities, each of which is represented by a value between 0 and 1.

An image-based dataset extracted from the ChaLearn dataset First Impressions is presented in the paper [11]. This dataset consists of selfies labeled with apparent personality traits. They sampled three or four frames from each video, yielding 30,935 images. Each image taken from the videos is cropped to resemble a selfie. Using OpenCV in Python, they performed face detection in each image and each image was then cropped so that the entire face was visible. Each image in the dataset was labeled with personality traits corresponding to the video from which it was sampled.

The authors of the paper [6] collected a self-introduction video dataset of 240 participants from the University of Chinese Academy of Sciences. Students in their undergraduate and graduate years made up the bulk of the participants. The participants were asked three questions related to introducing themselves, talking about their hometown, and their plans. To get the personality traits, they asked the participants to answer the BFI-44 questionnaire. This dataset is not public and permission is needed from the authors to access it.

The authors of the paper [12] built an end-to-end asynchronous video interview to detect the facial landmarks of the participants while recording the video. They built a website that allows the participants to record a video while answering predefined questions for the interview. The labels for the personality traits were annotated using the IPIP-50 personality test. The participants were asked to take the test after finishing the interview. This dataset consists of 120 samples and it is not publicly accessible and cannot be used by scientists.

None of the previous datasets had any information related to sales estimation and its relationship with personality traits. As such, there is a need to record a custom dataset that can be used for identifying and analyzing the relationship between personality traits and sales ability estimation. At the same time, this dataset gives general and valuable information in the field of personality analysis.

## 3. Dataset

In this section, we will delve into three aspects: data collection, data description, and data analysis. Data collection involves the process of gathering data from various sources. This includes self-evaluating surveys and self-interviews. Data description involves organizing and summarizing the collected data in a meaningful way. It includes describing the

characteristics of the dataset. Data analysis involves using statistical and other analytical techniques to conclude from the collected dataset.

### 3.1. Data Collection Methodology

In this subsection, we discuss the steps and methods used to collect the dataset. This includes information about the types of data sources and the procedures we followed to clean and preprocess the data. Additionally, we introduce the pipeline we used to move the data through various stages, such as data cleaning, formatting, and storage.

Stage 1: we developed a Google form to collect data from the participants and save it in our data storage. We asked the participants to record self-introduction videos while answering customized questions (the questions are shown in Table 1). Each participant used their device's camera (phone or laptop) and uploaded the video to the form. Then, we added a link that allows the participants to take the International Personality Item Pool (IPIP-120) and upload their personality traits results to the form [13]. We figured out that using this method for collecting data causes some difficulties, such as the format of the uploaded videos being different and needing more preprocessing work to prepare them to train the model.

**Table 1.** The interview questions that participants should answer while recording the video.

| Question | Description |
| --- | --- |
| 1 | Introduce yourself, such as your name, age, speciality, study, and a few pieces of information that express you. |
| 2 | What skills do you have that will help you work in a potential company? Where would you like to go to work? |
| 3 | Do you like to do sales? |
| 4 | Why do you want to do sales today? |
| 5 | What achievements are you proud of over the past 3 years? |

Stage 2: we also used a Google form to collect the data. We built a website using HTML, CSS, and Javascript to allow participants to record their self-introduction while answering our questions [14]. The website asks the users to access their device camera to initialize the process of recording. The participant has control to start and stop the recording process and also can swipe between the questions. Using this method, we came across the problem of multiple formats of the recorded videos by the participants, where all the videos are recorded in .webm format, which makes the pre-processing procedure easier. Using the website as a tool for recording videos is considered the most convenient and accessible way by the participant because it can run on all operating systems without any problems. A participant needs a browser and an internet connection to use the website. We re-implemented the IPIP-50 test in our form [15]. The participants have to answer 50 questions related to personality estimation. The participants have to choose an answer on a scale from 1 to 5, where 1 indicated very inaccurate and 5 indicated very accurate. The pipeline for collecting the dataset is shown in Figure 1.

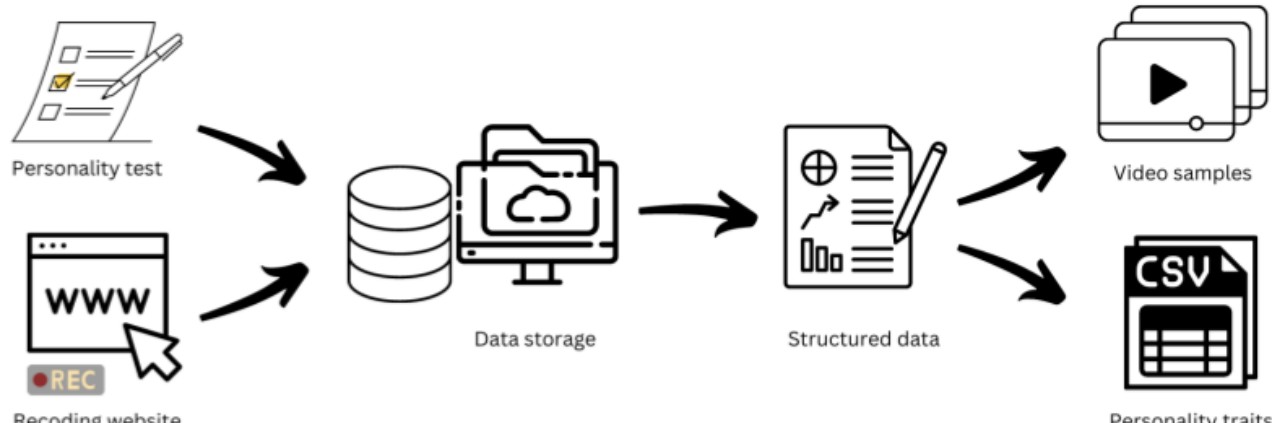

**Figure 1.** Data Collection Methodology.

### 3.2. Data Description

The collected data have the following information about the participants. They contain personality traits according to the OCEAN model, the participants' self-estimation for being good in sales, and our classification for the participants. Table 2 shows the data description. The classification is performed using a our developed deep learning model that analyzes the participant's video interview to estimate personality traits and uses expert-based knowledge to make the final classification depending on the extracted traits.

The participants' personality traits were evaluated by calculating their scores from the IPIP-50 they did after recording the self-introduction video. The participants evaluated their abilities to work in sales on a scale from 1 to 5, where 1 indicated poorly qualified and 5 indicated strongly qualified. Finally, we estimated the participants' ability to work in sales by our deep learning model that analyzes the video interview to extract personality traits, and then used our base knowledge to predict the classification score.

**Table 2.** Proposed dataset structure.

| Column Name | Description | Range |
| --- | --- | --- |
| Id_name | Unique id for each participant | - |
| Openness | Level of curiosity and independence | float: [0–1] |
| Conscientiousness | Level of reliability and perseverance | float: [0–1] |
| Extraversion | Level of friendliness and adventure | float: [0–1] |
| Agreeableness | Level of support and empathy | float: [0–1] |
| Neuroticism | Level of nervousness and anxiety | float: [0–1] |
| SE | Participants' self-estimation for their sales ability | integer [1–5] |
| OE | Our estimation for the sales ability of the participants | integer [1–10] |
| Participant video | 1–3 min video of participant answering interview questions | file |

A variety of samples make up the dataset that was gathered, and the bulk of the participants were students with a range of personality traits and sales experience. With an average duration of 120 s, the final dataset comprises 38 self-video interviews. The dataset contains different people of different races, nationalities (13.16% Syrians and 86.84% Russians), and gender (23.68% females and 76.31% males). The age range of the participants was between 19–46 years and the majority of the participants were in their twenties (according to a sales study in 2021 [16], this age range represents 28% of people who work in sales), which represents a big and active section of salespeople. The majority of the dataset were students with academic study (who are able to technically use the recording website and do the personality test). We also asked the participants to be honest and spontaneous as much as they could to guarantee the credibility of the videos.

To guarantee that the model will train correctly and that there would be no imbalance in any of the sample categories, we ensured that we sampled four video clips from each video participant and that these samples were distributed in the ratio 2:1:1 to the train, validation, and test sets. This ensures that the model trains normally and that there is no imbalance between any group of samples or any dominating characteristics in comparison to others.

Only high-quality gathered samples where the participant's face is apparent in the video were utilized in this investigation. Additionally, we used the samples where the subjects responded to our offered personality test and accepted our conditions for utilizing their videos in this study.

### 3.3. Data Analysis

We give an exploratory analysis of the data in this section. An Intel Core i5 personal computer with a 1.6 GHz processor, 16 GB of Memory, and an MX130i GPU was used for the data analysis presented below. We applied basic statistical analysis to the dataset. We calculated histograms of the personality traits (see Figure 2). The histograms of each personality trait are shown in Figure 3.

As we can see, the samples in the dataset are varied in each dimension, and the values are spread around the mean, forming a distribution similar to the normal distribution.

To extract the main statistical information about the dataset we used the "describe" method supported by the Python library "pandas", as shown in Table 3. We used the "seaborn" Python library to get the box plot representation of the data as shown in Figure 2. The box plot gives a graphical representation of the data through their quartiles. It also indicates where the majority of the data lies on each trait and identifies any outliers.

**Table 3.** Analyzing the personality traits.

|      | Extraversion | Agreeableness | Conscientiousness | Neuroticism | Openness |
| --- | --- | --- | --- | --- | --- |
| mean | 0.5718 | 0.6642 | 0.6605 | 0.4196 | 0.6713 |
| std  | 0.1719 | 0.1491 | 0.1470 | 0.1757 | 0.1364 |
| min  | 0.1750 | 0.2750 | 0.4000 | 0.0750 | 0.2900 |
| max  | 0.9000 | 0.9750 | 0.9500 | 0.7750 | 0.9750 |

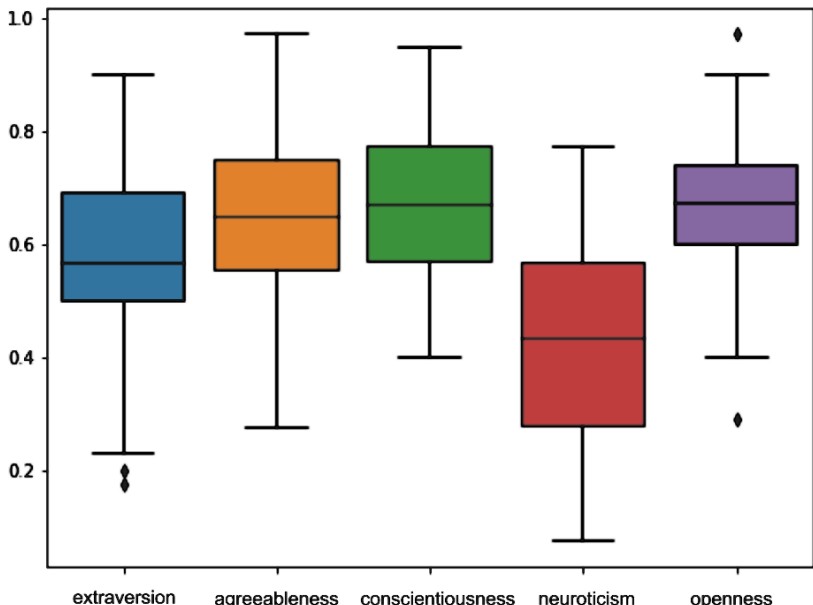

**Figure 2.** Box-plot representation for the personality traits distribution.

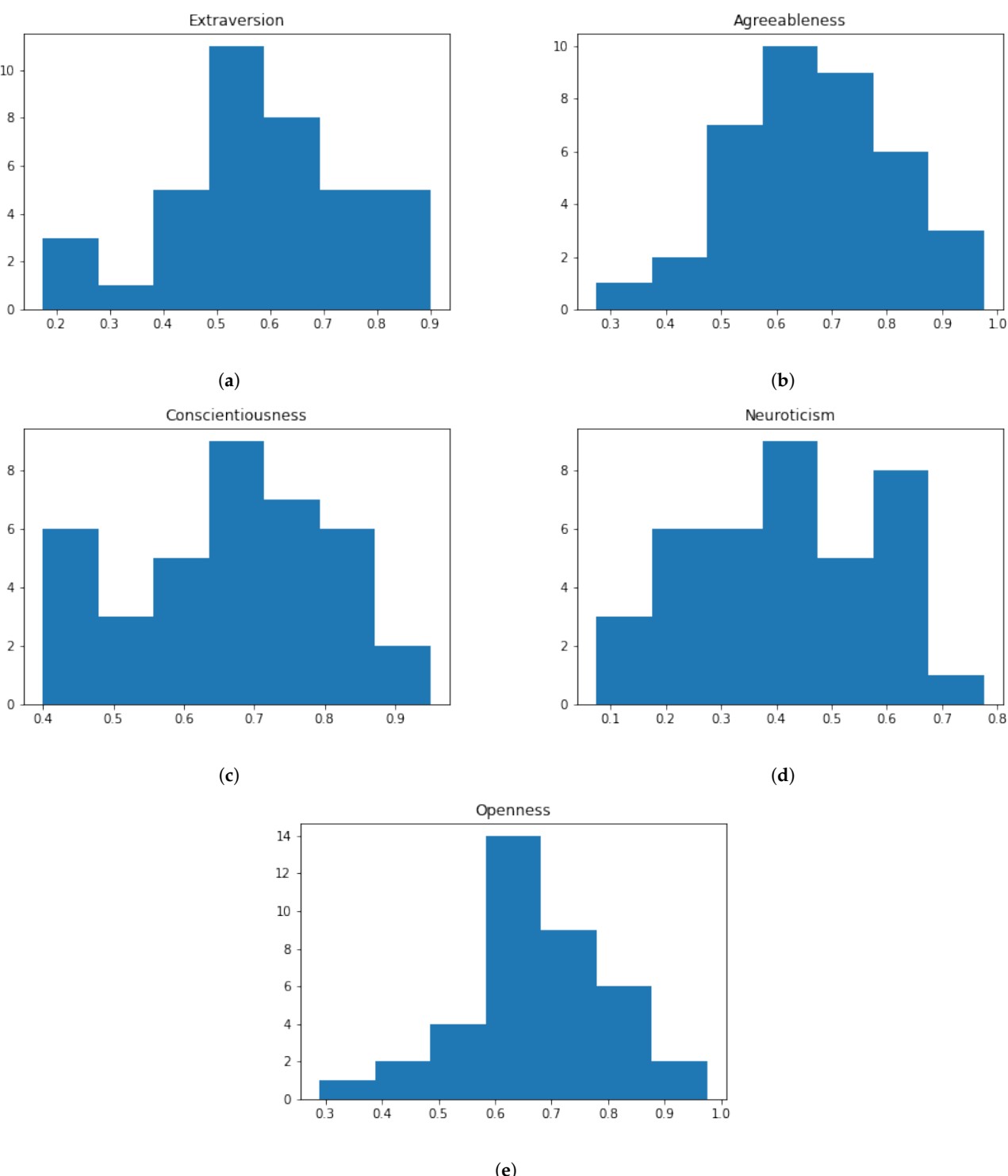

**Figure 3.** The histograms of the personality traits: (**a**) extraversion, (**b**) agreeableness, (**c**) conscientiousness, (**d**) neuroticism, (**e**) openness.

## 4. Data Evaluation

We applied the K-means clustering algorithm to cluster our dataset [17]. We chose the K-means clustering algorithm because it is suitable for small datasets and it iterates over all of the data points. We also wanted to cluster all the samples in the dataset without any outliers, so we used a centroid-based algorithm. We used personality traits (extraversion, conscientiousness, agreeableness, neuroticism, and openness) as features for the clustering algorithm. From the elbow chart shown in Figure 4, we figured out that the potential

number of clusters for our dataset could be two, three, or four clusters, as shown in Figure 5. To visualize the clustering results, we applied Principal Component Analysis (PCA) to transform the data into the 2D space [18].

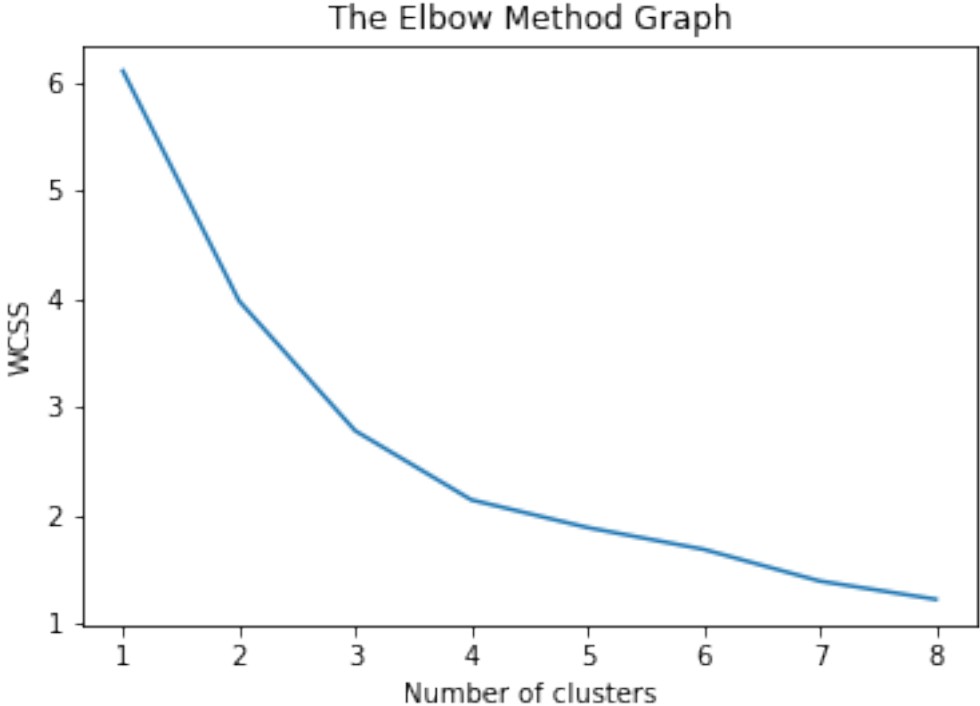

**Figure 4.** Elbow graph.

For each cluster, we counted the number of samples, as shown in Tables 4–6. Table 4 shows the number of samples in each resulting cluster after choosing K = 2, Table 5 shows the number of samples in each resulting cluster after choosing K = 3, and Table 6 shows the number of samples in each resulting cluster after choosing K = 4. This reflects how our dataset is split using the K-means clustering algorithm with a different number of clusters.

**Table 4.** Sample count for each of the two clusters.

| Cluster | Number of Samples |
|---|---|
| Cluster1 | 19 |
| Cluster2 | 19 |

**Table 5.** Sample count for each of the three clusters.

| Cluster | Number of Samples |
|---|---|
| Cluster1 | 7 |
| Cluster2 | 13 |
| Cluster3 | 18 |

**Table 6.** Sample count for each of the four clusters.

| Cluster | Number of Samples |
|---|---|
| Cluster1 | 6 |
| Cluster2 | 8 |
| Cluster3 | 12 |
| Cluster4 | 12 |

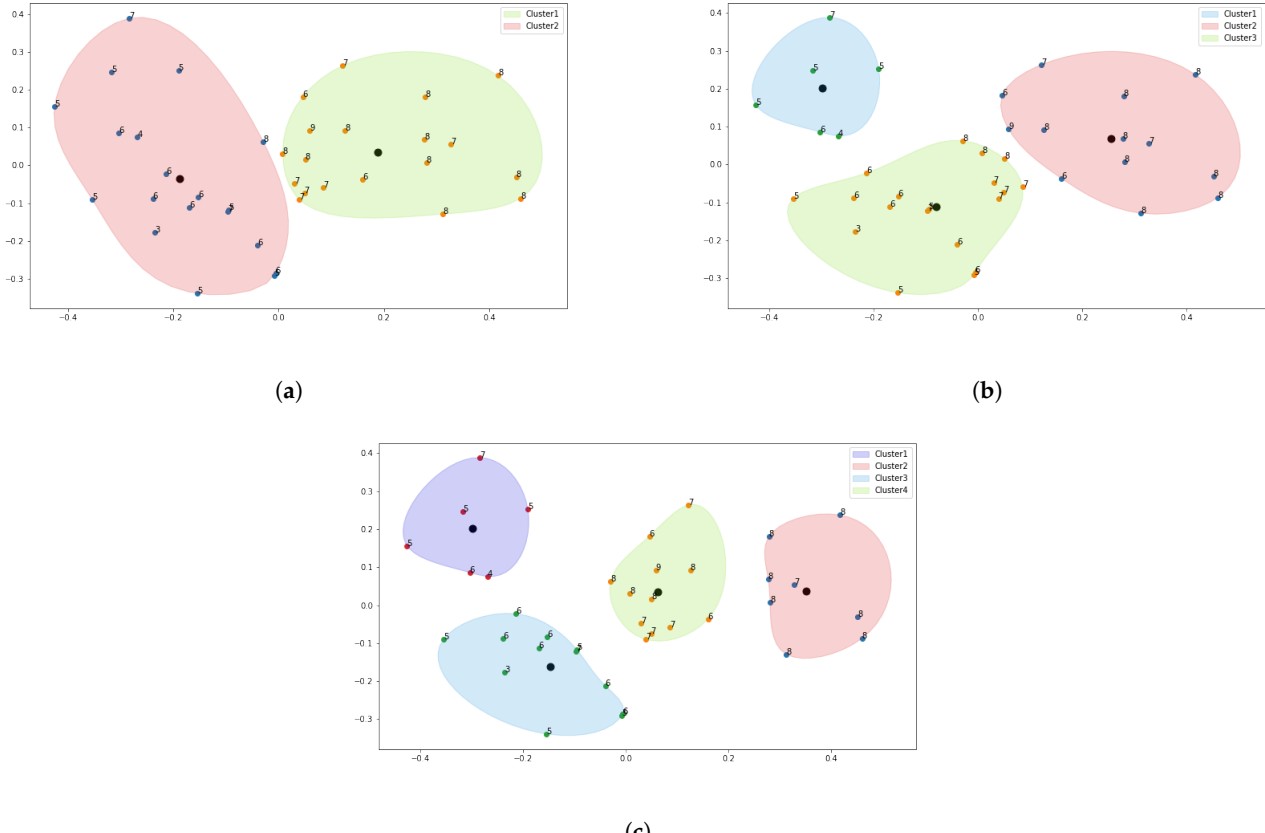

**Figure 5.** The results for clustering the dataset using K-means and after applying dimensionality reduction into 2D using PCA. (**a**) The clustering results in case of two clusters. (**b**) The clustering results in case of three clusters. (**c**) The clustering results in case of four clusters.

We also applied statistical analysis for each cluster, as shown in Tables 7–9. Table 7 shows the statistical key values (min, max, and mean) of the resulting clusters in case of choosing K = 2. The same applies for Tables 8 and 9 in case of choosing K = 3 and K = 4, respectively. The results of clustering the dataset are considered reasonable, as shown in "Human Sales Ability Estimation Based on Interview Video Analysis" [19], in that each cluster contains participants who have close ability potential for sales positions.

By examining the previous tables that summarized the valuable statistical information about the dataset and how the samples separated through the clusters, and also by analyzing the classification results for the participants' sales abilities depending on our previous study [19], we can conclude the classification state of each cluster as follows:

By examining Figure 5a, as well as Tables 4 and 7, which show the clustering results in the case of two clusters, we can state that Cluster1 represents samples with above-average sales abilities, whereas Cluster2 represents samples with below-average sales abilities.

**Table 7.** Analyzing the dataset in case of two clusters.

| #2 Clusters | | Extraversion | Agreeableness | Conscientiousness | Neuroticism | Openness |
|---|---|---|---|---|---|---|
| | Min | 0.4250 | 0.5200 | 0.4250 | 0.0750 | 0.6000 |
| Cluster1 | Max | 0.9000 | 0.9750 | 0.9250 | 0.4800 | 0.9750 |
| | Mean | 0.6808 | 0.7284 | 0.7211 | 0.3061 | 0.7384 |
| | Min | 0.1750 | 0.2750 | 0.4000 | 0.2500 | 0.2900 |
| Cluster2 | Max | 0.6600 | 0.8250 | 0.9500 | 0.7750 | 0.8500 |
| | Mean | 0.4629 | 0.6000 | 0.6000 | 0.5332 | 0.6042 |

**Table 8.** Analyzing the dataset in case of three clusters.

| #3 Clusters | | Extraversion | Agreeableness | Conscientiousness | Neuroticism | Openness |
|---|---|---|---|---|---|---|
| | Min | 0.1750 | 0.2750 | 0.4000 | 0.2500 | 0.2900 |
| Cluster1 | Max | 0.6000 | 0.6250 | 0.7250 | 0.5750 | 0.6000 |
| | Mean | 0.3629 | 0.5079 | 0.5179 | 0.4036 | 0.4879 |
| | Min | 0.4250 | 0.5200 | 0.4250 | 0.0750 | 0.6000 |
| Cluster2 | Max | 0.9000 | 0.9750 | 0.9250 | 0.4750 | 0.9750 |
| | Mean | 0.7200 | 0.7604 | 0.7088 | 0.2504 | 0.7615 |
| | Min | 0.3500 | 0.4500 | 0.4750 | 0.3600 | 0.5500 |
| Cluster3 | Max | 0.6700 | 0.8250 | 0.9500 | 0.7750 | 0.8500 |
| | Mean | 0.5461 | 0.6556 | 0.6811 | 0.5481 | 0.6775 |

**Table 9.** Analyzing the dataset in case of four clusters.

| #4 Clusters | | Extraversion | Agreeableness | Conscientiousness | Neuroticism | Openness |
|---|---|---|---|---|---|---|
| | Min | 0.1750 | 0.2750 | 0.4000 | 0.2500 | 0.2900 |
| Cluster1 | Max | 0.6000 | 0.6250 | 0.6500 | 0.5500 | 0.6000 |
| | Mean | 0.3900 | 0.5008 | 0.4833 | 0.3750 | 0.4692 |
| | Min | 0.7200 | 0.7000 | 0.6000 | 0.0750 | 0.6000 |
| Cluster2 | Max | 0.9000 | 0.9750 | 0.9250 | 0.4000 | 0.9750 |
| | Mean | 0.7869 | 0.8312 | 0.7369 | 0.2369 | 0.7844 |
| | Min | 0.2000 | 0.5100 | 0.4750 | 0.5500 | 0.5500 |
| Cluster3 | Max | 0.6100 | 0.8250 | 0.9500 | 0.7750 | 0.8500 |
| | Mean | 0.4829 | 0.6621 | 0.6600 | 0.6192 | 0.6562 |
| | Min | 0.4250 | 0.4500 | 0.4250 | 0.0750 | 0.6000 |
| Cluster4 | Max | 0.8000 | 0.7500 | 0.8500 | 0.4800 | 0.8750 |
| | Mean | 0.6083 | 0.6367 | 0.6988 | 0.3642 | 0.7121 |

By examining Figure 5b, as well as Tables 5 and 8, which show the clustering results in the case of three clusters, we can state that Cluster1, Cluster2, and Cluster3, represent samples with poor, good, and satisfactory sales abilities, respectively.

Finally, by examining Figure 5c, as well as Tables 6 and 9, which show the clustering results in the case of four clusters, we can state that Cluster1, Cluster2, Cluster3, and Cluster4 represent samples with poor, very good, normal, and good sales abilities, respectively.

## 5. Discussion

In this paper, we introduced a video-based dataset for personality trait estimation. We explained our data collection methodology and the techniques we used to gather this dataset. We also explored the samples in the datasets by running a statistical analysis. This study has some limitations related to the honesty of the participants and how honest and reliable they were while answering the questions.

The first limitation is a subjective estimation of personality traits (if all the participants tricked us into not answering the questionnaires honestly or they just randomly chose the answers).

The second limitation is related to how strictly the participants followed the instructions (if a participant saw the interview questions on the website in advance). These limitations might affect the reliability of the dataset and the accuracy of the ground truth for the personality traits.

The third limitation is related to the fact that some people could hide their emotions to deal with the fact that we asked the participants to be honest and spontaneous as much as they could. However, we tried to make the recording process similar to the real-life

interview where the participants answered the questions one by one without any previous knowledge of the questions. Additionally, the majority of the participants were students and what was important for the study was to be able to use the website to record the self-interview while expressing yourself spontaneously.

We understand these limitations and we asked the participants to follow the instructions for recording videos. We explained and clarified to them how important it is to give reliable and honest answers to the questions. Finally, our evaluation shows that, in general, these limitations did not occur.

### 6. Conclusions

In this paper, we introduced our human face video dataset for personality traits detection (VDPT). The dataset contains 38 video samples for different people of different races, nationalities (13.16% Syrians and 86.84% Russians), and genders (23.68% females and 76.31% males). The bulk of the samples were students with a range of personality traits and sales experience. The dataset contains self-video interviews for the participants recorded through our website while they answered our pre-defined questions. The dataset provides the personality traits of each participant according to the OCEAN model, the self-estimation for sales ability, and our classification of the participants' sales abilities.

The dataset can be used for analyzing video interviews to extract and study the personality traits described by the OCEAN model. It could also be used to assess the sales abilities of a person by studying and analyzing their video interview and to study the relationship between personality traits and people who are suitable for sales positions. The collected dataset is considered a good resource for training models related to job hiring and understanding the human personality. It is also considered valuable for studying personality traits through video data, especially since only a few video-based datasets are available for this kind of study, not to mention that collecting video-based datasets is assumed to be a difficult task.

We explained our data collection methodology and the techniques we used to gather the dataset. We introduced our method for clustering the dataset using K-means clustering and analyzing the clustering results. We also discussed the final results and the limitations of our study.

Future work will focus on expanding the dataset to include larger amounts of samples and collecting more information from the participants related to the sales field.

**Author Contributions:** K.K. was in charge of creating the datasets, developing the website, creating and modifying the forms, analyzing the datasets, and writing the article. A.K. was responsible for conceptualization, peer review. A.M. was responsible for financing the research. D.Z. was responsible for finding the participants for experiments and conceptualization. All authors have read and agreed to the published version of the manuscript.

**Funding:** This work was supported by the Russian State Research FFZF-2022-0005. Data evaluation (Section 4) has been supported by Russian Foundation for Basic Research project #19-29-06099.

**Institutional Review Board Statement:** Not applicable.

**Informed Consent Statement:** Informed consent was obtained from all subjects involved in the study.

**Data Availability Statement:** Our dataset is available through the following link: https://doi.org/10.5281/zenodo.8068262.

**Conflicts of Interest:** The authors declare no conflict of interest.

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
