# Peer review of "VPTD: Human Face Video Dataset for Personality Traits Detection"

_data, 2023_

Round 1

Reviewer 1 Report

In this study, 38 video samples for different race, nationality and gender were examined by using the human face video dataset to determine personality traits. The dataset also provides the opportunity for participants registered via the website to make their own video calls while answering our predefined questions. includes. The dataset can be used to analyze video calls to extract and examine the personality traits described by the OCEAN model. It can also be used to assess a person's sales abilities by reviewing and analyzing a video call, and to examine the relationship between personality traits and people suitable for sales positions. The method of clustering the dataset and analyzing the clustering results using K-means clustering is introduced. Study results are not satisfactory. The tables given are not explained in detail. Studies done in the abstract and conclusion section should be presented in summary. Academic language should be regulated. The intro part should be rewritten using more detailed references.

In this study, 38 video samples for different race, nationality and gender were examined by using the human face video dataset to determine personality traits. The dataset also provides the opportunity for participants registered via the website to make their own video calls while answering our predefined questions. includes. The dataset can be used to analyze video calls to extract and examine the personality traits described by the OCEAN model. It can also be used to assess a person's sales abilities by reviewing and analyzing a video call, and to examine the relationship between personality traits and people suitable for sales positions. The method of clustering the dataset and analyzing the clustering results using K-means clustering is introduced. Study results are not satisfactory. The tables given are not explained in detail. Studies done in the abstract and conclusion section should be presented in summary. Academic language should be regulated. The intro part should be rewritten using more detailed references.

Author Response

Dear reviwer,

Thank you for valuable feedback to our paper that aloow us to enhance it.

Attached you can see the detailed responce.

Authors.

Reviewer 2 Report

The article proposes the creation of a dataset where it relates personality traits with collected videos. The article is well written and well structured.

The authors argue that they have "participants with different nationalities, genders, and races.” In my opinion the volume of data is very small (38 videos) and has little diversity which can be biased: only 23,68% are female and most of the participants are Russians (86,84%). Speaking of diversity of participants. What is the age range of the participants?

Will it be important for the study? And academic training (schooling), is it important? Some people are trained, or educated, to hide their emotions. Other people lie constantly. How does the dataset react to this?

In the end, I didn't understand the relationship between the collected data and the person's sales ability. I think this needs to be better explained.

Minor issue: In the last paragraph of the introduction the words “The rest of the paper …” are repeated.

English is good.

Minor issue: In the last paragraph of the introduction the words “The rest of the paper …” are repeated.

Author Response

(The authors gave the same response as above.)

Reviewer 3 Report

1) Please include the results in Introduction section. 

2) The authors only provided the data. What is the main contribution of this study? 

3) What are the limitations of this study?

4) The authors stated that "Where the participants answered 33 interview questions related to sales." What are these interview questions?

5) No information is given about participants. Please provide demographic information about participants.

6) How did the authors decide to apply K-means clustering algorithm? Maybe other clustering algorithms have better performance. 

7) Please provide more details about Tables 4,5,6,7,8, and 9. 

8) There is no discussion section in this article. Please discuss your results in detail. 

Moderate editing of English language is required.

Author Response

(The authors gave the same response as above.)

Round 2

Reviewer 1 Report

The study is acceptable as it is.

Reviewer 2 Report

The little diversity presented in the people who performed the test (gender, age, etc.) causes the results to be biased, which reduces the significance of the results.

Reviewer 3 Report

The authors answered all my queries and modified the manuscript accordingly. The paper is now acceptable. 

Minor editing of English language is required.